# SpecSeg Network for Specular Highlight Detection and Segmentation in Real-World Images

**DOI:** 10.3390/s22176552

**Published:** 2022-08-30

**Authors:** Atif Anwer, Samia Ainouz, Mohamad Naufal Mohamad Saad, Syed Saad Azhar Ali, Fabrice Meriaudeau

**Affiliations:** 1Laboratoire d’Informatique, du Traitement de l’Information et des Systèmes (LITIS), Normandie Université, UNIROUEN, UNIHAVRE, INSA Rouen, 76000 Rouen, France; 2Centre for Intelligent Signal& Imaging Research (CISIR), Electrical and Electronic Engineering Department, Universiti Teknologi PETRONAS, Seri Iskandar 32610, Malaysia; 3Aerospace Engineering Department, King Fahd University of Petroleum & Minerals, Dhahran 31261, Saudi Arabia; 4ImViA, Université Bourgogne-Franche-Comté, 71200 Le Creusot, France

**Keywords:** specular highlights, image segmentation

## Abstract

Specular highlights detection and removal in images is a fundamental yet non-trivial problem of interest. Most modern techniques proposed are inadequate at dealing with real-world images taken under uncontrolled conditions with the presence of complex textures, multiple objects, and bright colours, resulting in reduced accuracy and false positives. To detect specular pixels in a wide variety of real-world images independent of the number, colour, or type of illuminating source, we propose an efficient Specular Segmentation (SpecSeg) network based on the U-net architecture that is expeditious to train on nominal-sized datasets. The proposed network can detect pixels strongly affected by specular highlights with a high degree of precision, as shown by comparison with the state-of-the-art methods. The technique proposed is trained on publicly available datasets and tested using a large selection of real-world images with highly encouraging results.

## 1. Introduction to Specular Highlights

Regions with specular reflections in an image are generally unwanted yet mostly unavoidable features. This is why the problem of specular highlight detection is challenging and has been an area of progressive research for both traditional photography and digital imaging since the beginning. Specular reflections are extremely hard to avoid in real-world conditions since they depend on several factors, including variables related to the illuminating source as well as the target object in the scene. These factors include azimuthal, zenith orientations of the illuminating source and the object as the primary factor in the presence of specular highlight, alongside factors such as the material of the surfaces interacting with the light. In most natural-world conditions, one or more of these factors are uncontrollable, which makes the presence of specular reflections impossible to avoid. Specular highlights are a highly informative feature and have an essential role in image processing and computer graphics. Specular reflections are essential for human vision as they provide powerful visual cues about the shape of the objects, the material of the object, and the location of the illuminating light source. However, apart from specific applications, specular reflection is generally considered an undesirable feature in the image processing domain, causing loss of chromatic and textural information that is often vital to applications [1].

While there are several physical models for the definition of specular reflections in images, the most prominent one is the Dichromatic Reflection Model (DRM) proposed by Shafer et al. [2]. They proposed method involves decomposing an image into specular and diffuse images as defined by the linearly additive DRM. The incoming luminance is divided into two components, body reflectance and interface reflectance. The body reflectance. is part of the visible spectrum of wavelengths that are reflected after interacting with the particles of the body below the surface and represents the colour of the target body. The interface reflectance is the part of the wavelength that is reflected directly from the surface and represents the illuminant’s colour. The DRM is inherently an under-determined system with a non-trivial solution for the separation of the two components of an image, as defined by the following equation:(1)$$L\left(\lambda ,i,e,g\right)\;=\;m_{d}\left(i,e,g\right)c_{d}\left(\lambda \right)\;+\;m_{s}\left(i,e,g\right)c_{s}\left(\lambda \right)$$ where *d*, *s* stand for the diffuse (body) and specular (interface) components, respectively, *c* is the spectral power distribution, $m_{d}$, $m_{s}$ are the geometric scale factors. The light wavelength is denoted by λ and i,e,g are angles of the incident light, emitted light, and phase angle (with respect to the surface normal), respectively. A matte surface is comprised mostly of the body reflectance, whereas specular surfaces contain a combination of both spectral and diffuse components. Furthermore, the probabilistic independence of specular and diffuse highlight is not constant as it depends on whether the surface is textured or smooth, as shown in Figure 1.

The DRM model is valid for inhomogeneous materials only [2], which are materials of uniform composition throughout and cannot be mechanically separated into different materials. The model is based on three core assumptions. Firstly, the reflection from the surface is invariant with respect to rotation around the surface-normal, and there are no inter-reflections among surfaces. Secondly, the body reflection is Lambertian which means that the brightness is independent of the viewing direction. Lambertian models describe a perfectly diffuse surface that scatters incident illumination isotropically (equally in all directions) independent of the viewer’s position. Although this reflection model is not physically plausible, it is a reasonable approximation to many real-world surfaces such as matte paint. Lastly, the specular reflection has the same colour as the illumination and tends to be polarised. While most assumptions seem to limit the model’s applicability to real-world problems, they allow the model’s generalisation and increase its applicability to a wide assortment of problems. Thus even after several ideal assumptions, the DRM model has broad applicability to understanding and mitigating specular highlights.

In digital imaging, the requirement of detecting the occurrence of specular reflections and identifying all the corresponding affected pixels accurately becomes an essential yet formidable task. The affected specular pixels have to be segmented before any processing algorithm can be used to remove the specularity and mitigate the undesirable effects of the reflection. As the DRM model is an ill-posed problem with more unknown than known variables, accurately segmenting and detecting specular pixels in an image is challenging. A single image does not provide enough information regarding the physical orientation of the light source or the surface orientation required to calculate the surface normals about which light is specularly reflected. Since specular pixels are generally represented by the brightest pixels in an area, they are hard to differentiate from lighter colours in the scene. This is further accentuated with the presence of large brightly-lit regions, such as the sky, or the presence of any light source directly in the image. Similarly, pixels nearing the colour of the light source are hard to differentiate from specular reflections, thus making the accurate segmentation of specular pixels a highly arduous task. Furthermore, since the strength of the illumination is captured as pixel intensity, specular pixels are represented by higher intensity values, often near or fully saturated values (i.e., (255,255,255) in a standard RGB image). The saturation of pixels means that the object’s colour, texture, and other spatial information encompassed by the specular pixels is lost. In order to recover this lost information, robust mitigation of the specular pixels is required so that the underlying features such as colour, texture, etc., can be estimated correctly. Sensor-clipping due to over-pixel exposure from strong specular reflection also results in loss of image information. This further complicates the problem and requires detailed and intelligent methods of accurate specular detection in images.

Several real-world examples of images containing varying amounts of specular pixels are shown in Figure 2. As can be seen, the shape, size, area and locations of specular regions vary widely depending on multiple conditions in which the image is taken. Segmenting specular pixels is a gruelling task for manual annotations, and it is even more challenging to automate it by computer vision algorithms.

## 2. A Survey of Specular Highlight Detection and Segmentation in Literature

As has already been established, specular highlights and reflections lower the visibility and clarity of the contents of the images, affecting the results of other algorithms, such as segmentation and classification, causing them to fail. Hence, while being an ill-posed problem, reflection removal is one of the most challenging topics in image processing. Over the years, the problem of detecting specular pixels and the affected areas has been attempted using handcrafted and predetermined techniques, falling under the classical techniques. Recently, machine learning-based solutions have seen significant growth, with promising results primarily from deep-learning-based solutions. In the following sections, we go over an in-depth literature review of the previously proposed solutions.

### 2.1. Classical Specular Detection and Segmentation Methods

Specular highlight segmentation has proven to be extremely challenging since it is an ill-posed problem. While specular reflections are easily distinguishable by human vision, it is a tough ask for digital image processing systems. Traditional techniques have always been based on simplifying the problem in some manner, including assumptions regarding the colour of light, the transmission medium and its refractive index, the object’s material, etc. While most assumptions are valid for solving a problem, they are mostly unrealistic and do not represent an accurate real-world scenario. The DRM model has proven to be a reasonably accurate model to explain the causes of specular reflections and thus forms the basis of a large selection of detection and mitigation techniques. The subsequent sections review the most used methods and techniques proposed by research works over the years.

#### 2.1.1. Segmenting Specular Highlights Using Chromaticity

Shafer et al. [2] were the first to propose the Dichromatic reflection model, which became the fundamental model for understanding and explaining nearly all reflection models. Their breakthrough paper used the spectral distribution of light and its colour coordinates to identify and separate the colour pixels into diffuse and specular components. Unlike previous models such as the Phong, [3] which uses specific reflectance functions to predict the reflection amount, DRM is based on the physical model of reflection, making it more intuitive and realistic. Klinker et al. [1] based their work on DRM and showed that the colour histogram of an image forms a T-shaped distribution with uniform diffuse regions. Using geometric heuristics instead of colour information, they estimate a single global diffuse colour, which can be extended to several segmented regions of homogeneous diffuse colour and estimate the body and reflection components. Klinker and Shafer et al. [4] also proposed modelling of highlights as a linear combination of both surface and body reflections and modelled camera properties to account for camera limitations and showed that generating the intrinsic images from a single image was possible. Several other methods using colour space transformations [5,6,7] were also proposed to segment out the specular pixels. However, the assumption of pure white global illumination and uniformly single-coloured, non-textured objects in most of these methods limited the application. Tan and Ikeuchi et al. [8] proposed a method based on the difference in logarithmic differentiation of the normalised input and specular-free images. Yoon et al. [9] were the first to introduce the two-band specular free image obtained by subtracting the minimum of the three RGB channel values from each pixel. These values are then compared to neighbour intensity ratios to their corresponding ratios in the specular-free representation for separating highlight pixels. Shen et al. [10] later modified the PSF image by Yoon et al. to make its chromaticity robust to noise by adding an offset factor and solving the DRM equation as a least-square problem for mixed specular-diffuse regions. Later, several other approaches [11,12,13] built upon the PSF image approach with varying results.

#### 2.1.2. Polarisation, Low-Rank Approximations, and Other Approaches

The concept of polarisation is directly related to the problem of specular highlight segmentation due to the highly polarised nature of specular reflections [14]. Due to this, significant research has been done on segmenting and removing specular highlights in images using polarisation. It is noteworthy that the significance of DRM is further increased since it can be used in conjunction with the polarised nature of specular reflection to explain its occurrence and mitigation. An example of how polarised and unpolarised specular reflections are affected by a polariser filter can be seen in Figure 3. Wolf et al. [15] were one of the earliest groups to use polariser images to classify materials in images using the Fresnel reflection model. They monitored the variation of light by capturing multiple images while rotating the polariser filter in front of a camera and noted that the brightness of diffuse materials varied as the polariser was rotated. They also noted that the variation between the minimum and maximum intensity captured fluctuates in a sinusoidal pattern as a function of the polariser angle. Nayar et al. [16] were one of the first to simultaneously use polarisation and colour information to separate the diffuse and specular reflection components by capturing at least six images at different polar angles. They used polarisation to acquire independent local estimates of the colour of the specular component, forcing each image pixel to lie in a linear colour subspace and then thresholding it to achieve the desired separation. Kim et al. [17] extended Nayar et al.’s work by dividing the colour space into a specular line-space and a diffuse plane space. The diffuse pixels are selected by thresholding the intensity variation while rotating the polariser. The spatial variation in the specular components is then smoothed out using an energy function. Umeyama et al. [14] applied Independent Component Analysis (ICA) to images captured through a rotating polariser to separate the diffuse and specular components. More recently, Wen et al. [18] proposed a polarisation-guided model that can be used to cluster pixels with similar diffuse colours. They formulated the problem in an optimised global energy minimisation function, resulting in specular reflection separation in images.

Over the years, one of the popular methods of solving the specular reflection problem has been to treat it as noise in an image and utilise techniques that can mitigate the effect of noise in images. By assuming specular reflections as noise, authors have shown that methods such as noise filtering, low-rank approximations [19,20,21], blind source separation [22], and other minimisation [23] techniques can be used to approximate the image data, freeing it from the effects of noise. An alternate approach to treating specular reflections is to capture multiple images from different angles by taking multiple images [24] using light field cameras [25,26] or hyperspectral cameras [27] that specialise in taking multi-focal but spatially coherent images. As can be seen, there has been a significant amount of research over the years, and multiple ways and techniques have been attempted to segment out the damaged pixels in an image. A summary of the classical methods for specular highlight segmentation is given in Table 1.

### 2.2. Deep Learning Based Methods

While there have been many studies of specular highlight detection over the years, most classical methods conduct a visual evaluation on a few selected images, mostly without annotated ground truth or highlight masks. This has led to a very unrealistic quantitative evaluation of highlight detection algorithms on real-world images where the lighting can vary significantly from ideal conditions. Specular reflections caused by inter-reflections between objects or light reflecting off other surfaces in the scene cause multiple issues, often not addressed by classical methods. During the last decade, the benefits of machine learning have become quite evident with a substantial impact, especially in image processing. Furthermore, deep learning has seen a significant amount of growth and development not only in the core techniques but also in frameworks for implementing efficient and robust deep-learning implementations.

Several solutions have been proposed in recent years to accurately identify the specular pixels in medical images by leveraging machine learning algorithms. Sanchez et al. [44] used a two-stage segmentation and classification approach to identify specular regions in colonoscopic images and then filtered through a linear SVM classifier. Akbari et al. [45] utilised an adaptation between RGB and HSV colour spaces using a non-linear SVM classifier and then inpainted the detected regions. One of the earliest methods toward a more generalised and smart specular highlight detection method was proposed by Lee et al. [46] which implemented detection of specular reflections by a single layer perceptron. Looking forward to the state-of-the-art deep learning methods, Funke et al. [47] were the first to utilize a Cycle Generative Adversarial Network (CycleGAN) to localize specular regions for endoscopic images. Their method used data with weak labels indicating the presence or absence of specular highlight in a training image only.

It is worth mentioning that typically most of the state-of-the-art deep learning-based methods are geared towards training the network to *mitigate* the specular highlights using supervised or unsupervised training methods. This means that very few works exclusively focused on deep learning methods to detect specular pixels, which is the focus of this work. One of the recent papers that focused on detecting specular highlights in real-world images was proposed by Fu et al. [48]. The proposed Specular Highlight Detection network SHDNet used multi-scale contrast features to detect specular pixels that are scale agnostic. SHDNET uses a convenient and embeds a multi-scale context contrasted feature network for successfully detecting specular highlights in real-world images. The authors also present a large-scale dataset of roughly real-world images, which include manually annotated highlight regions. In addition to the primary dataset, they also prepared a testing dataset of 500 images in the wild called the WHU-TRIW dataset. Fu et al. [49] proposed another large-scale dataset comprising 16k real-world images alongside a multi-task network for Joint Specular Highlight Detection and Removal (JSHDR). They propose a Dilated Spatial Contextual Feature Aggregation (DSCFA) to detect and accurately remove highlights of varying sizes. A comprehensive list of the relevant machine learning-based methods on real-world images for detecting specular highlights in images is presented in Table 2.

### 2.3. Limitations of the Current State-of-the-Art

The accurate detection of specular highlights is significant in many applications. Classical methods for accurately detecting specular highlights have difficulty detecting pixels accurately in a wide variety of scenes containing lighter coloured objects, bright backgrounds, or complex-shaped objects with irregular specular reflections. One of the significant issues faced by the classical techniques is the robustness and generalisation of the techniques. While the methodologies are based on firm mathematical foundations and optimisation techniques, they are primarily based on assumptions that significantly limit their applications to general real-world images that are not part of their dataset. Thus, while the results are significantly better on the selected set of images, they do not apply to any general image taken from a generic camera under uncontrolled settings. Multiple research works on treating specular reflections using colour space transformations attempted to understand and tackle the problem purely from an objective often tested on a minimal set of images which fails to work beyond their preferred set. Methods based on polarisation classically use a manual polariser filter that is rotated to acquire images at different polarimetric angles. This means that the images are temporally incoherent, and unless taken of a static object under a static and controlled environment, the images face alignment issues where pixels do not share the same spatial instance between the polar images. This also limits the number of images that can be acquired as a significant amount of effort is required to take a broad and generalised dataset. Several assumptions are also made for classical methods to work, which are sometimes not reflective of real-world conditions, e.g., a single illumination is mostly assumed with a non-existent or minimal amount of inter-reflections from surrounding surfaces. The illuminants selected are assumed to be of pure white colour with known spectral power distribution (SPD) to simplify all chromaticity-based methods. It is further assumed that each segmented cluster has uniform diffuse chromaticity. While being very helpful for modelling the problem of specular highlight, these and other assumptions do not reflect real-world images’ randomness and limit the generalisation and applicability of methods.

Since most limitations are not considered for deep learning-based methods, it is quite clear that modern state-of-the-art methods are significantly more robust and can cater to a much more comprehensive range of images. However, limitations are enhanced in the presence of outdoor images, which have both strong illumination and inter-reflections in an uncontrolled and often stochastic environment. Outdoor environments have illumination from the sun as an omnidirectional light source, causing light to bounce off in often undesirable directions and strength. Strong light sources also result in more significant specular regions in images, which makes the regions easily visible but also easily confused with the objects in the scene, as well as causing a significant loss of information in the area, which hinders the recovery of colour and other information in the affected region.

## 3. Specular Highlight Segmentation Network (SpecSeg Network)

As already detailed in the preceding section, accurate detection and segmentation of specular pixels from real-world images have significant implications in various fields. This work intends to fill the research gap and add to the current state of the art in specular highlight segmentation. To achieve this, we propose a specular highlight segmentation network that is simple to model, fast to train, and works on images used in the literature as well as a wide variety of general real-world images. Most state-of-the-art deep learning models are structurally complex, with a complex organisation of deep hidden layers and innovative, unique features such as attention and other methods. Secondly, due to their complex design, they require a significant time to train and fine-tune due to there being many hyperparameters in the model and deep neural network layer structure. This, in turn, causes significant hindrances in research and development due to unoptimised training times required while expected nominal results are not achieved. Furthermore, complex and deep networks also mandate the utilisation of expensive and powerful hardware, consuming much power while training and re-training. We avoid both these pitfalls by our proposed Specular Highlight Segmentation Network (SpecSeg Network for short), based on the proven U-net model, which is a highly reliable yet straightforward model that was initially proposed for medical segmentation [51]. Our experiments show that this decision makes the specular highlight detection network simple to build and requires significantly less time and fewer resources to train. This enables increased experimentation and re-training opportunities without trading accuracy or precision from the existing state-of-the-art methods. Furthermore, we also show that by using the SpecSeg Network it is possible to detect specular highlights after fast training on a relatively small dataset and generate accurate detection results on real-world images. The affected pixels are accurately marked in a wide assortment of images taken in random uncontrolled settings and improve upon the existing state-of-the-art methods in specular highlight detection.

### 3.1. SpecSeg Network Model and Implementation

Since its inception, U-Net has proven to be a breakthrough for segmentation tasks and has been instrumental in paving the way for developing a more advanced encoder–decoder style of network. The network is named after the U-shape of the hidden layers, combining an encoder–decoder arrangement for downsampling the input to a bottleneck and upsampling again to an output image, with convolution, activation, and pooling operations between its successive hidden layers. Skip connections allow the network to propagate context information from higher resolution layers to the decoder’s generated outputs and significantly affect the quality and accuracy of the Unet output [52]. By parsing the input image through down convolutions and pooling in an encoder, the network learns to identify the target regions in a scale-agnostic manner. The network thus learns to segment images in an end-to-end setting, i.e., the network input is a raw image (which can be in a single or multi-channel colour space), and the output image is in the form of a segmentation map. The u-net network has been shown to work with high accuracy and detect objects with substantial shape variations, weak borders and inset or overlapping objects. Due to these properties, the u-net forms our developed SpecSeg Network’s primary building block for detecting specular highlights in real-world images. The proposed deep convolutional network layout is shown in Figure 4, and the following sections discuss the design and reasons for selecting the hyper-parameters.

#### 3.1.1. Encoder and Decoder Blocks

SpecSeg comprises five encoder blocks and four decoder blocks based on the classical U-net pattern, and each path from the encoder is passed to the decoder via a skip connection. Each encoder block consists of two 2D convolutional layers with filters (k)=3 and stride (s)=3 with ‘same’ padding and uses ReLU activation in the output of each convolutional layer. The original proposed U-net configuration has inspired the (3×3) filter. However, a stride of (3×3) is added to avoid overlap when convolving the filter, as it was experimentally determined to give the most favourable results during testing and evaluation. While in the original paper, Ronneberger et al. [51] propose unpadded convolutions in the encoder section, it has been shown [53] that the choice of padding has a direct effect on the performance of a model. Without padding, the input layer volume size reduces too quickly as a deeper network is designed. Stacking multiple unpadded layers also ignores the image’s border pixels, resulting in a loss of learnable information around the borders. Since specular highlights can also extend to the border of the input images, adding padding around the border increases the chances of detecting specular pixels near the border of the input image.

An incremental dropout of 10%,20%, and 30% respectively is also introduced between the two convolutional layers of the first, third, and fifth encoder block to improve the robustness of the learned features. By incrementally increasing the dropout, the network learns sparser representations of the high-level features and improves the accuracy of the detection of specular pixels. The training was done on a batch size of 16, and a Batch Normalisation (BN) layer was introduced in the encoder sections before the pooling layer. BN has proven to be a reliable normalisation method for segmentation networks [54], and the same was confirmed by our experimentation, making it a sound choice. Lastly, to reduce the variance and computational complexity as we go deeper into the u-net, we need to reduce the size of the feature map. This is achieved with a MaxPooling layer that selects the maximum value out of a 2×2 block, reducing the size of the feature set. Maxpooling ensures that only the most critical features (denoted by the maximum valued pixels) are taken from each block.

The decoder block mostly mirrors the encoder block setup defined above with a few notable changes. Firstly the decoder performs an upscaling operation. This is done using 2D transpose convolutional layers with filters k=2 and stride s=2. A similar incremental dropout between two consecutive convolutional layers is also used. However, the final convolutional layer uses a filter and strides of k=1,s=1 respectively and sigmoid activation to generate a 256×256×16 mask images of the entire batch similar in size the input images.

Thus the overall U-net structure takes batches of 16 images of resolution 256×256 as input and generates mask images as output for all 16 images while learning the weights during the downscaling–upscaling operations in the encoder-decoder pairs.

#### 3.1.2. Loss Functions

For deep learning problems, loss functions depend profoundly on the problem being solved and are often tailored to the task at hand. For specular highlight segmentation, we selected a linear combination of Dice similarity coefficient (DSC) [55] and Focal loss [56] as experiments proved that the combination of these losses showed the best segmentation results.

Dice similarity coefficient (DSC) is a spatial overlap index developed to measure the pixel-level similarity between two images, where one is generally the binary mask image. DSC loss function has values in the range of 0–1. Lower values indicate minimum spatial overlap between two sets of binary segmentation results, whereas larger values nearing 1 indicate increasing overlap, where 1 represents 100% complete overlap. The dice similarity coefficient has been widely adopted in biomedical segmentation problems where manually annotated lesions or cancerous cell datasets are available to train segmentation algorithms. Mathematically, the dice similarity loss ($L_{Dice}$) is defined as (Equation 2).
(2)$$L_{Dice}\left(p,\hat{p}\right)=1-\frac{2\sum p\hat{p}}{\sum p+\sum \hat{p}}$$
where *p* is the ground truth, $\hat{p}$ is the predicted probability and
$$p\in \left\{0,1\right\}\;,\;0\leq \hat{p}\leq 1$$

Focal loss [56] addresses class imbalance during training by applying a modulating term to the cross entropy loss to focus learning on hard misclassified samples. Alternatively, it can be visualised as a dynamically-scaled cross-entropy loss, where the scaling factor decays to zero as confidence in the correct class increases. Intuitively, this scaling factor (γ) automatically down-weights the contribution of more accessible training samples and rapidly converges the model to focus on more challenging examples. Mathematically focal loss ($L_{Focal}$) can be defined as:(3)$$L_{Focal}\left(p,\hat{p}\right)=-\alpha {\left(1-\hat{p}\right)}^{\gamma }p\log\left(\hat{p}\right)-\alpha \left(1-p\right){\hat{p}}^{\gamma }\log\left(1-\hat{p}\right)$$

By adding the losses mentioned above, we can create a total loss that calculates the true positive segmented pixels and enables the network to focus on the misclassified samples of the training dataset. The dice loss maximises the overlap between predicted and actual labels, whereas the focal loss addresses class imbalance by reducing the effect of biased or skewed classification on the predicted results. The total loss function is defined as a linear combination of both the Dice loss and Focal loss and is used for backpropagating over all learnable parameters.
(4)$$L_{Total}=L_{Dice}+L_{Focal}$$

SpecSeg network is implemented using Tensorflow 2.8’s sequential API as it provides easy and high-performance execution of the relatively straightforward network. SpecSeg is optimised using ADAM optimiser with $\beta_{1}=0.9$ and $\beta_{2}=0.999$. A batch size of 16 was used for training the network for 200 epochs. The training was stopped after 200 epochs to avoid overfitting, as the validation curve was seen to flatten out, indicating that further training might lead to a poorer generalisation learned by the network [54]. All training and testing for SpecSeg network were done on the Nvidia P100 card, released in April 2016 and based on Nvidia’s proprietary Pascal Architecture. Several datasets with specular masks are available publicly for testing as detailed in the Table 3 but for testing and comparison, two of the most recent datasets were used; namely Whu-Specular dataset [57] and SHIQ dataset [49]. The datasets were split into train and validation sets in 90%, 10% ratio, respectively, whereas the initially provided test sets with each dataset were used for testing. The qualitative and quantitative results are discussed in the following subsections in detail. The segmented specular highlight results are compared with state-of-the-art specular detection methods provided in the literature. Unfortunately, most competing methods do not provide network implementations or pre-trained weights, which limits direct retraining or testing on customised datasets. Therefore the testing was done on the same datasets and similar architecture to ensure a fair comparison to the available competing results.

## 4. Results

The results of segmenting specular highlights using SpecSeg network on the Whu-Specular dataset [57] are shown in Figure 5 and on the SIHQ dataset [49] in Figure 6. The input image is in the top row, followed by the specular masks in the second row as given in the datasets. The last row is the predicted specular pixels from our SpecSeg network. Results on images used in the existing literature as well as self-acquired images are presented in Figure 7 with additional segmentation results attached as Appendix A. A zoomed-in view of the predicted specular segmentation on select images is shown in Figure 8 in order to highlight some key observations in the discussion Section 5. The training and inference time of SpecSeg was also compared to other networks to show the performance benefits gained due to the reduced complexity of the proposed network.

The quantitative results of the testing done on the datasets used are presented in Table 4. The quantitative comparison was done using three metrics, S-measure (S-m) [62], F-measure (meanF), and MAE. Several segmentation methods evaluated by Fu et al. [48] have been directly included here from their works for a broader comparison. In their paper, all learning-based methods were re-trained on the same dataset (WHU-Specular dataset), and the authors fine-tuned the hyperparameters to give the best possible results. SpecSeg was also trained on the same training dataset, and the same validation and test sets were used to generate a fair comparison.

Outdoor conditions present the most significant and extreme challenge for any specular highlight detection network. The presence of a bright sky and strong sunlight in unpredictable imaging conditions and scene contents present the biggest failure challenges for all algorithms. The proposed SpecSeg network was tested on acquired outdoor images taken on a clear and bright sunny day. Some of the segmentation results are presented in Figure 9. Lastly, ablation studies to explore the effects of various hyper-parameters and loss functions are also presented in Section 5.2.

## 5. Discussions

Visually comparing with the manually annotated masks in Figure 5 and Figure 6, we can see that the network can detect all specular regions and generate masks closely resembling the ground-truth images. The detection of specular regions is valid for various materials in the images, including plastic, wood, metallic, and ceramic objects of irregular shape. Even small specular regions in the images are detected quite accurately. The images are taken under natural lighting conditions and have an unknown number and orientation of light sources. This results in specular pixels of various intensities and colours depending on the illuminating source colour. Specular highlights on light-coloured surfaces are also detected accurately, which is often hard for most conventional algorithms. Note that the manually annotated masks result from human visual interpretation of specular pixels in an image and are therefore susceptible to misrepresentation, especially around the region borders. While the highly saturated pixels are easy to identify and mark, the distinction becomes significantly challenging and blurry around the edges of the specular region, where the falloff to diffuse colour can be soft enough such that some pixels may be wrongly marked as specular and vice versa. This is challenging in real-world images because there are multiple light sources in various orientations and of different strengths. As opposed to medical image masks, where there is a single illumination positioned nearly concentric with the camera for acquiring endoscopic and colonoscopic images, resulting in very sharp specular boundaries that medical experts can mark, resulting in the masks being highly accurate, making the qualitative analysis easier and quantitative analysis more meaningful. Despite these shortcomings, the manually annotated masks provided are an excellent baseline for evaluating all qualitative and quantitative segmentation methods. Looking at a few segmentation results more closely in image Figure 8, we can see that SpecSeg network is successfully able to detect regions that are on light-coloured objects (a), small in size (b), in multiple blocks with cavities inside specular regions, (c) clipped around the edges of the image, (d) and most importantly can detect specularity correctly from images on a white background (e). As can be seen in the Figure 8c, non-specular regions surrounded by specular pixels are accurately detected despite the small size. Specular regions that are along the image edges such as Figure 8d are also accurately detected without any problem. Additionally, almost all classical segmentation methods are unable to distinguish white backgrounds in images from specular pixels (Figure 8e) and are often some of the most challenging images to segment out for SOTA algorithms. SpecSeg is able to perform reliably in all these unique conditions. The results of the segmentation masks generated by SpecSeg are significantly better than the classical methods, as seen from the quantitative results presented in Table 4. The results are also comparable to other state-of-the-art deep learning-based methods. SpecSeg can achieve a higher MAE score while getting close and comparable results for S-measure and F-Measure to Fu et al.’s [49] SHDNet. As seen by the statistical summary of the entire test dataset shown in Figure 10a the scores are within a tightly bound distribution with only a couple of outlier cases. Figure 10b presents the training and validation curves.

Owing to several challenges, as discussed in Section 3.1.2, there is a significant lack of specular datasets containing images taken outdoors in bright sunny conditions with specular pixel annotations or ground truth diffuse images. Therefore, training a specular segmentation network with large amounts of outdoor images is impossible. As shown in Figure 9, specular regions are detected reasonably well despite the presence of bright sky areas and intense reflections. The sky and water puddles are not falsely detected as specular regions, nor are large white regions on road signs or car bodies. As expected, there are a few challenges, and specular reflection detection can be improved on outdoor images. There are no ground truth diffuse images or specular annotations publicly available to analyse the results quantitatively. However, to our knowledge, this work is the first to present an accurate specular highlight detection network that works on indoor as well as outdoor images with reasonably accurate results on the latter, despite there being no availability of any large outdoor specular dataset available to train the network.

### 5.1. Performance Comparison

One of the most significant caveats of deep learning is the significantly staggeringly large times required for training the networks. To compare training time with the other methods, our proposed network was trained on the Whu-specular training dataset for 200 epochs for a mere 40 min on a P100 (Pascal architecture). In comparison, the SHDNet achieved its results after training for 100 epochs in 80 h on a GTX-1080Ti (also Pascal architecture). This significantly reduces training time without the need for additional computational power to achieve comparable segmentation results. For training and inference comparison, Fu et al. [48] trained and tested their network on the NVIDIA GeForce GTX 1080Ti, which was released in March 2017 and is based on the Pascal Architecture by Nvidia. in comparison, our training and testing were done on the NVIDIA P100, released in April 2016 and also based on Pascal Architecture. Having the same architecture helps to maintain similarity in the performance, allowing the computation of performance metrics to be as close as possible. Note that the authors of SHDNet have not provided their PyTorch or Tensorflow implementation code for public access, so retraining their network on any dataset was impossible. It is clear from the results in Table 5 that the training time required by SpecSeg is an order of magnitude better than all other competing networks. Furthermore, the inference time is also faster than the competing networks. As noted above, since the code or training weights of SHDNet or JSHDR have not been provided publicly, it was impossible to retrain and test on the same hardware for a 100% fair comparison. However, the hardware used is comparable and can be treated as similar for all intents and purposes for deep neural network training.

### 5.2. Ablation Studies

In order to test the proposed network, an in-depth ablation study was carried out by varying different aspects of the network. As shown by the performance comparison in Table 5, the training time for the network is very low, which significantly helps in testing different configurations and hyper-parameter tuning of the network. Several variations were constructed by editing the activation functions of the SpecSeg network. A separate training session also noted the benefit of using batch normalisation. Additionally, varying the loss functions with alternate losses versus the proposed joint loss $L_{Total}$ was also studied. A comparison of different metrics calculated from the resulting ablation studies is shown in the Table 6.

The proposed losses combined with batch normalisation and LRelu activation give the best results for PSNR, MSE, Dice, and S-measure scores, whereas the SSIM score is lower only by a negligible amount. Using Leaky ReLU activation gives the overall best scores as it avoids the vanishing gradient problem. The combination of dice and focal losses appear to converge successfully towards the best results on the test dataset. All ablation tests were carried out on the same hardware and did not see any change in training time.

## 6. Conclusions and Future Work

A deep-learning-based method for segmenting specular highlights from single images was presented. The proposed network is significantly fast to train with limited images and accurately detects specular reflections in real-world images with no restriction on illumination conditions for image acquisition. The novelty of the proposed method is the customisation and tailoring of an established architecture for the accurate detection of specular highlights that have not been applied to the segmentation of specular highlights to real-world images in the prior literature. Furthermore, the proposed network’s training time requirement and inference performance are significantly better than other competing networks trained and tested on comparable hardware. The segmented specular highlights are comparable with state-of-the-art specular detection methods provided in the literature. Unfortunately, as most competing methods do not provide their implementations or pre-trained weights, direct retraining or testing on custom datasets is impossible. We also show that the proposed network can detect specular highlights in outdoor images taken under extremely bright conditions, with good results. To our knowledge, no other prior work has presented a specular highlight detection network that works on indoor and outdoor images with reasonably accurate results on both conditions. The proposed methods can further be improved by possibly incorporating self-attention methods to improve the robustness of the learned specular features. In the future, we intend to improve the inference process’s accuracy and implement recovery of the affected textural information for dovetailing into advanced pipelines for specular highlight mitigation. We also intend to further improve the results on outdoor images by acquiring more extensive amounts of outdoor data with specular highlights using polarimetric cameras and training SpecSeg for improved segmentation results.

## Figures and Tables

**Figure 1 sensors-22-06552-f001:**
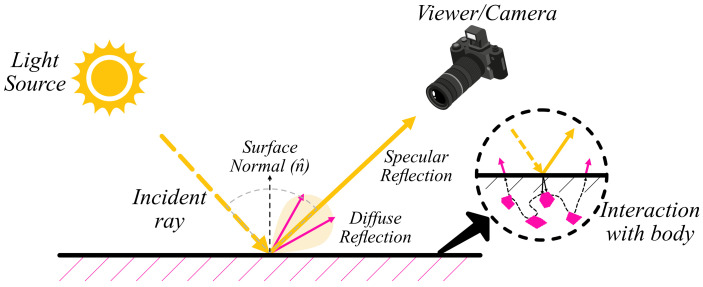
The dichromatic reflection model for inhomogeneous materials.

**Figure 2 sensors-22-06552-f002:**
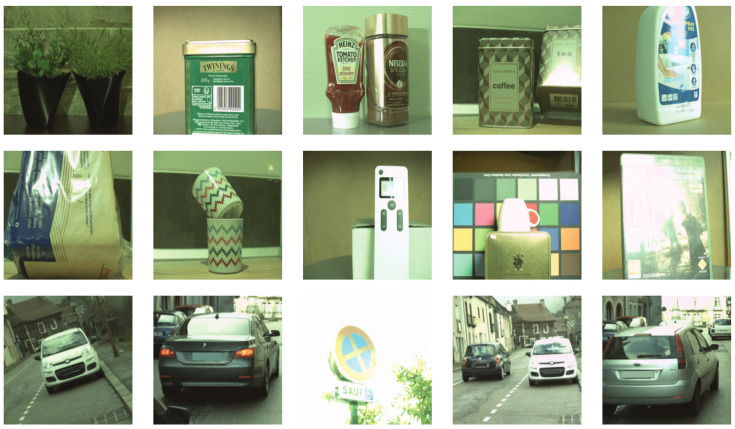
Real-world examples of specular reflection in images.

**Figure 3 sensors-22-06552-f003:**
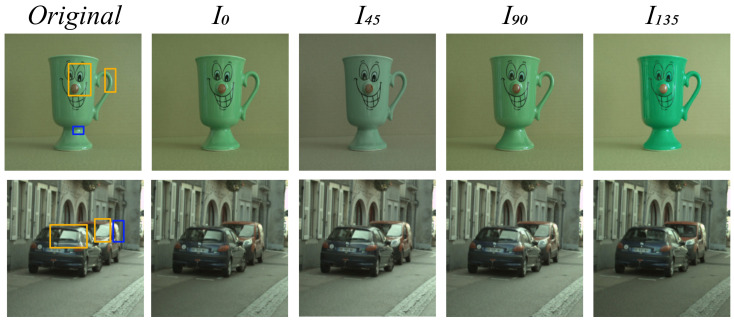
Variation in specularity with the variation of polarisation angle (orange areas) in uncontrolled environments. Note that unpolarised light causes specular reflection regardless of polarisation filter angle (blue areas).

**Figure 4 sensors-22-06552-f004:**
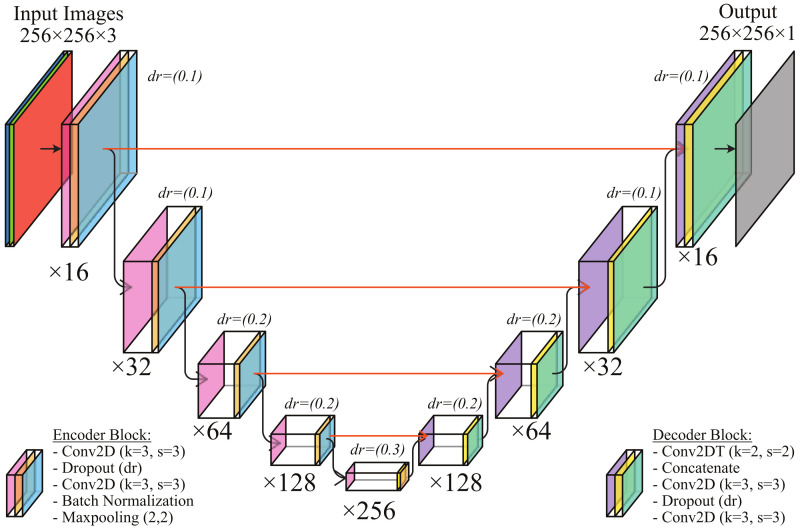
SpecSeg configuration based on the U-net architechture.

**Figure 5 sensors-22-06552-f005:**
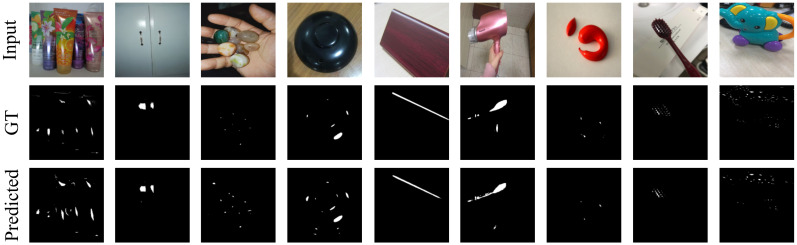
Segmentation results of SpecSeg network as compared to manually labelled ground truths in the Whu-Specular dataset [57].

**Figure 6 sensors-22-06552-f006:**
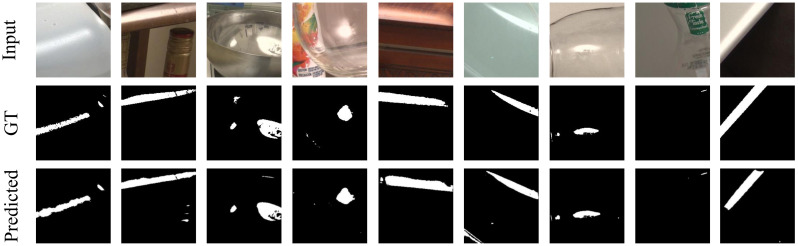
Segmentation results of SpecSeg network as compared to manually labelled Ground Truths (GT) in the SIHQ dataset [49].

**Figure 7 sensors-22-06552-f007:**
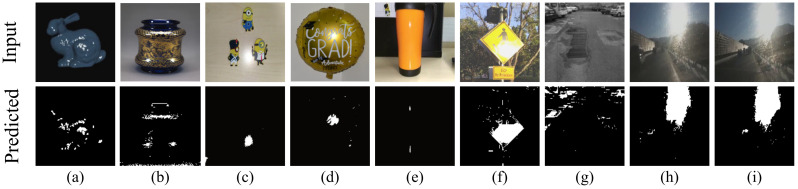
Segmentation results of SpecSeg on real world images from various sources and self acquired images. Sub-images from left to right (**a**) generated from [63], (**b**) image from [64], (**c**–**g**) taken by authors, (**h**,**i**) video frames from iRoads dataset [65].

**Figure 8 sensors-22-06552-f008:**
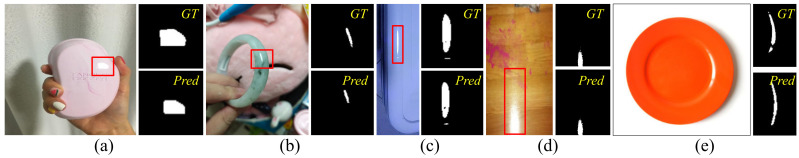
Zoomed-in ground truth (GT) and prediction (Pred) views of the marked sections in RGB images from [57]. SpecSeg network is successfully able to detect regions that are (**a**) on light-coloured objects, (**b**) small in size, (**c**) in multiple blocks with cavities inside specular regions, (**d**) clipped around the edges of the image, (**e**) detect specularity correctly from images on a white background.

**Figure 9 sensors-22-06552-f009:**
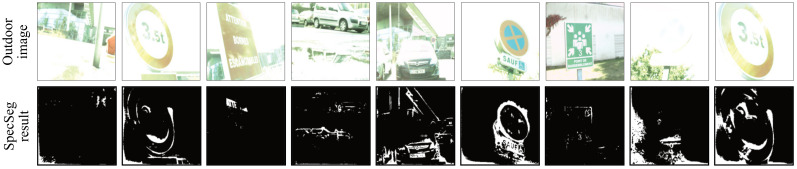
Specular segmentation results on outdoor images acquired on a sunny day and under clear sky conditions. Specular reflections detected under extreme conditions are plausible and significantly better than any other state-of-the-art technique. Note that brightly lit regions such as the sky or water puddles are not detected as specular regions.

**Figure 10 sensors-22-06552-f010:**
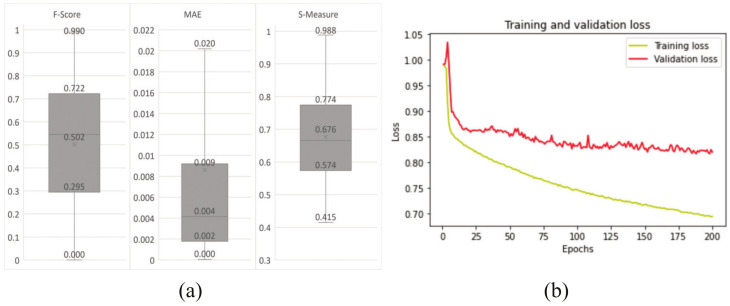
(**a**) A summary of the metrics over the entire dataset. (**b**) Training and validation losses after 200 epochs. The training was stopped after 200 epochs to avoid overfitting by the network.

**Table 1 sensors-22-06552-t001:** Summary of major non-deep learning based methods for specular highlight segmentation.

Name	Year	Category ^1^^,^^2^	Technique	Color Space
Bajcsy et al. [6]	1996	Segmentation	Segmentation by Hue, Saturation	S-space
Umeyama et al. [14]	2004	Separation	Polarisation, ICA	Greyscale
Tan et al. [8]	2005	Separation	Chromaticity, Colour Spaces	RGB
Tan et al. [28]	2006	Separation	Spatial Colour Distributions	RGB
Shen et al. [10]	2008	Separation	Chromaticity based	RGB
Shen et al. [11]	2009	Separation	Pixel clustering	RGBD
Mesloushi et al. [29]	2011	Segmentation	Chromaticity	CIE XYZ
Yang et al. [30]	2013	Separation	Region growing algorithm	HSI
Kim et al. [31]	2013	Segmentation	Dark Channel Prior	RGB
Zou et al. [32]	2013	Segmentation	Dark Channel Prior	RGB
Akashi et al. [19]	2016	Segmentation	NMF	RGB
Shah et al. [33]	2017	Segmentation	SIFT in sequential images	RGB
Yamamoto et al. [34]	2017	Separation	SVD, Energy minimisation	RGB
Alsaleh et al. [35]	2019	Separation	Low-Rank Temporal Data	RGB
Fu et al. [23]	2019	Separation	Optimisation	RGB
Li et al. [36]	2020	Separation	RPCA	RGB
Son et al. [37]	2020	Separation	convex optimisation	RGB
Ramos et al. [38]	2021	Separation	histogram matching	YCbCr
Haefner et al. [39]	2021	Separation	HDR Imaging for separation	RGB
Bonekamp et al. [40]	2021	Separation	Multi-Image Optimisation	RGB
Kim et al. [41]	2021	Segmentation	Geometric estimation	RGB
Ramos et al. [38]	2021	Separation	histogram matching	YCbCr
Tominaga et al. [42]	2021	Segmentation	Iterative estimation process	RGB
Wen et al. [18]	2021	Separation	Polarisation	RGB
Li Furukawa [43]	2022	Separation	RPCA, Photometric Stereo	RGB

^1^ Separation: Methods that separate distinct specular and diffuse images that are additive. ^2^ Segmentation: Methods that segment out specular pixels from the original image, but do not generate diffuse image.

**Table 2 sensors-22-06552-t002:** Summary of the machine learning based methods for specular highlight segmentation in real-world images.

Name	Year	Category ^1^^,^^2^	Application ^3^	Architecture	Loss Functions	Evaluation
Lee et al. [46]	2010	Segmentation, Mitigation	Real-world	Single layer perceptron	-	-
Sanchez et al. [44]	2017	Segmentation	MIS	SVM	-	DICE
Akbari et al. [45]	2018	Segmentation	MIS	SVM	-	DICE, Specificity, Precision
Funke et al. [47]	2018	Segmentation, Mitigation	MIS	SpecGAN	Cyclic loss	MSE PSNR, SSIM
Fu et al. [48]	2020	Segmentation	Real-world	SHDNet	BCE, IOUE	F-measure, MAE, S-measure
Fu et al. [49]	2021	Segmentation, Mitigation	Real-world	JSHDR	BCE, L2	Accuracy, BER
Monkam et al. [50]	2021	Segmentation, Mitigation	MIS	Scaled-UNet, GatedResUNet	Mask loss, Valid loss, Perceptual loss, Style loss, Total variation loss	SNR, DICE, SSIM, IoU

^1^ Mitigation: Methods that generate specular-free images. ^2^ Segmentation: Methods that segment out specular pixels from the original image, but do not generate diffuse image. ^3^ Application: Real-world images or Medical Imaging Systems (MIS).

**Table 3 sensors-22-06552-t003:** List of specular imaging datasets.

Dataset Name	Year	Category	Total Images	Specular Mask	Diffuse Image	Test-Train Split	Size
Spec-DB [58]	2003	Real-world	300	**✓**	**✓**	**✗**	10 MB
CVC-ClinicDB [59]	2015	Medical Imaging	612	**✓**	**✗**	**✗**	263 MB
CVC-ClinicSpec [44]	2017	Medical Imaging	59	**✓**	**✗**	**✗**	6 MB
Whu Specular [48] ^1^	2020	Real-world	4310	**✓**	**✗**	**✓**	2 GB
PolaBot [60]	2020	Real-world	177	**✓**	**✗**	**✗**	584 MB
Specular Highlight Image Quadruples (SHIQ) [49] ^2^	2021	Real-world	16,000	**✓**	**✗**	**✓**	10.8 GB
2022 SIHR [61]	2022	Real-world	200	**✗**	**✓**	**✓**	503 MB

^1^ Dataset used for training and evaluation in this work. ^2^ Dataset used for evaluation in this work.

**Table 4 sensors-22-06552-t004:** Qualitative comparisonof SpecSeg network to classical and deep learning SOTA methods.

Metrics	Year	Type	S-m^1^	meanF ^1^	MAE ^2^
Tchoulack et al. [66]	2008	Classical	0.132	0.027	0.423
Chen et al. [67]	2018	Deep learning	0.619	0.451	0.019
Zhang et al. [68]	2019	Classical	0.521	0.410	0.021
Hou et al. [69]	2019	Classical	0.491	0.218	0.053
Zheng et al. [70]	2019	Deep learning	0.480	0.202	0.049
Hu et al. [71]	2020	Deep learning	0.412	0.108	0.091
Fu et al. [48]	2020	Deep learning	0.793	0.676	0.006
SpecSeg	2022	Deep Learning	0.676	0.502	0.008

^1^ Higher is better. ^2^ Lower is better.

**Table 5 sensors-22-06552-t005:** Training and inference time comparison of different segmentation networks.

Author	Network	GPU	Epochs	Training Time	Inference Time
Monkam et al. [50]	ScaledUNet	GTX 2080Ti	50	-	3.43 ms
Ronneberger et al. [51]	UI-Net	NVidia Titan	-	10 h	14.13
Fu et al. [48]	SHDNet	GTX1080Ti	100	80 h	-
Fu et al. [49]	JSHDR	GTX 2080Ti	100	3 days	-
Ours	SpecSeg	Nvidia P100	140	40 min	3.1 ms

**Table 6 sensors-22-06552-t006:** Ablation study results of different variations of the SpecSeg network.

	PSNR ↑	SSIM ↑	MSE ↓	F_m ↑	S_m ↑
No BN	23.7213	0.9539	0.0092	0.4662	0.6643
BN + SparseCE loss	25.2064	0.9628	0.0076	0.5072	0.6598
Elu Activation	21.9828	0.9494	0.0122	0.4308	0.6138
Linear Activation	24.0415	0.9609	0.0092	0.0067	0.5214
Baseline (BN + LReLU + $L_{Total}$)	25.2211	0.9625	0.0073	0.5278	0.6761

## Data Availability

Publicly available datasets were utilized in this study. (1) The Whu Specular dataset [48] can be found at the authors Github repository https://github.com/fu123456/SHDNet, accessed on 27 July 2022. (2) The SHIQ dataset [49] can be found at the authors Github repository https://github.com/fu123456/SHIQ, accessed on 27 July 2022. (3) The proposed SpecSeg network implementation presented in this work is available at author’s Github repository https://github.com/Atif-Anwer/SpecSeg, accessed on 27 July 2022.

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
