# Peer review of "SpecSeg Network for Specular Highlight Detection and Segmentation in Real-World Images"

_sensors, 2022, doi:10.3390/s22176552_

Round 1
Reviewer 1 Report
a) The activation function ReLu is used. Is it possible to compare the different activation functions against the linear function? And other more?
b) Show the results, the error graphs and the precision.
c) It is recommended to divide the results and discussion section into two different sections. In the discussion, it is recommended to address the proposed method's advantages and disadvantages versus state-of-the-art.
d) The images shown in the results seem to be taken indoors. The outdoor environment has too many variables to be considered. It is recommended to test or show evidence of the performance of the proposed methodology using datasets with outdoor objects.
e) Show in table 3 the dataset used.
f) Although the results show the efficiency of the neural network for Specular Highlight Detection, why not consider complementing the work and filling the gaps left in the images? Show complete images without Specular Highlights.
Author Response
Response to Reviewer 1 Comments
Point 1: The activation function ReLu is used. Is it possible to compare the different activation functions against the linear function? And other more?
Response 1: Ablation study on different activation functions, batch normalization and training losses added in section 5.2. Additional qualitative results and quantitative results of an ablation study were also added including various activation functions, normalization and losses.
Point 2: Show the results, the error graphs and the precision.
Response 2: Training and validation graphs have been added.
Point 3: It is recommended to divide the results and discussion section into two different sections. In the discussion, it is recommended to address the proposed method's advantages and disadvantages versus state-of-the-art.
Response 3: Paper restructured as suggested with individual results and discussion sections.
Point 4: The images shown in the results seem to be taken indoors. The outdoor environment has too many variables to be considered. It is recommended to test or show evidence of the performance of the proposed methodology using datasets with outdoor objects.
Response 4: Results of segmentation on self-acquired outdoor images added.
Point 5: Show in table 3 the dataset used.
Response 5: Dataset used for training and testing is now indicated in table 3.
Point 6: Although the results show the efficiency of the neural network for Specular Highlight Detection, why not consider complementing the work and filling the gaps left in the images? Show complete images without Specular Highlights.
Response 6: The research carried out by the authors consists of two separate parts. The first is the accurate detection of specular highlights and second is the mitigation of the detected highlights, each performed by a separate network. The performance gains by the training and inference time with the proposed specular detection network are noteworthy as compared to the state-of-the-art and have implications in the proper mitigation of specular highlights. Furthermore, a draft publication on an adversarial network for mitigation of specular highlights is already under review process.
Reviewer 2 Report
Overall, the results of this paper and the research methods are reasonable, and there are no major problems with the structure of the paper.
The main problem is such results are relatively backward in compare to the current publications; if specular highlight detection can only detect regions, and the performance is presented in terms of the consistency of the detection results and ground truth.
Please refer to [51] and [52] cited in this article. The author introduced strategies for patching and mitigating highlight areas in follow-up studies the following year. Even considering only the performance of detection, this paper does not make much progress compared to [51] (published in 2020).
If this paper can add reduction and correction for the detected specular highlight area, it will be very helpful in the field of medical imaging, such as the correction of endoscopic images.
At the same time, in the introduction part, some of the techniques discussed in the past research are irrelevant to this article. The focus of this paper is on using deep learning to detect specular highlights, so other techniques, such as applying optical techniques for detection is irrevelent here. It is suggested to be deleted or reduced to focus on the discussion.
Author Response
Response to Reviewer 2 Comments
Point 1: Overall, the results of this paper and the research methods are reasonable, and there are no major problems with the structure of the paper. The main problem is such results are relatively backward in compare to the current publications; if specular highlight detection can only detect regions, and the performance is presented in terms of the consistency of the detection results and ground truth. Please refer to [51] and [52] cited in this article. The author introduced strategies for patching and mitigating highlight areas in follow-up studies the following year. Even considering only the performance of detection, this paper does not make much progress compared to [51] (published in 2020). If this paper can add reduction and correction for the detected specular highlight area, it will be very helpful in the field of medical imaging, such as the correction of endoscopic images.
Response 1: The research carried out by the authors consists of two separate parts. The first is the accurate detection of specular highlights and second is the mitigation of the detected highlights, each performed by a separate network. A draft publication on the network for mitigation of specular highlights by an adversarial network is already under review process.
The focus of the current research work is currently not towards medical imaging. But for future work, a dedicated SpecSeg network trained on medical images will be explored based on the suggestions by the esteemed reviewers..
Point 2: At the same time, in the introduction part, some of the techniques discussed in the past research are irrelevant to this article. The focus of this paper is on using deep learning to detect specular highlights, so other techniques, such as applying optical techniques for detection is irrevelent here. It is suggested to be deleted or reduced to focus on the discussion..
Response 2: Reduced and streamlined related literature on classical methods. Reduced the tabular summary on classical methods for correspondence to the paper's deep learning method as suggested.